# Thermosensitive Polymer-Modified Mesoporous Silica for pH and Temperature-Responsive Drug Delivery

**DOI:** 10.3390/pharmaceutics15030795

**Published:** 2023-02-28

**Authors:** Kokila Thirupathi, Madhappan Santhamoorthy, Sivaprakasam Radhakrishnan, Selvakumari Ulagesan, Taek-Jeong Nam, Thi Tuong Vy Phan, Seong-Cheol Kim

**Affiliations:** 1Department of Physics, Government Arts and Science College for Women, Karimangalam, Dharmapuri 635111, Tamil Nadu, India; 2School of Chemical Engineering, Yeungnam University, Gyeongsan 38541, Republic of Korea; 3Department of Organic Materials and Fiber Engineering, Jeonbuk National University, 567 Baekje-daero, Deokjin-gu, Jeonju-si 54896, Republic of Korea; 4Division of Fisheries Life Sciences, Pukyong National University, Nam-gu, Busan 48513, Republic of Korea; 5Institute of Fisheries Sciences, Pukyong National University, Gijang-gun, Busan 46041, Republic of Korea; 6Center for Advanced Chemistry, Institute of Research and Development, Duy Tan University, 03 Quang Trung, Hai Chau, Danang 550000, Vietnam; 7Faculty of Environmental and Chemical Engineering, Duy Tan University, 03 Quang Trung, Hai Chau, Danang 550000, Vietnam

**Keywords:** mesoporous silica, drug delivery, thermosensitive copolymer, pH-stimuli, biocompatibility

## Abstract

A mesoporous silica-based drug delivery system (MS@PNIPAm-PAAm NPs) was synthesized by conjugating the PNIPAm-PAAm copolymer onto the mesoporous silica (MS) surface as a gatekeeper that responds to temperature and pH changes. The drug delivery studies are carried out in vitro at different pH (7.4, 6.5, and 5.0) and temperatures (such as 25 °C and 42 °C, respectively). The surface conjugated copolymer (PNIPAm-PAAm) acts as a gatekeeper below the lower critical solution temperature (LCST) (<32 °C) and as a collapsed globule structure above LCST (>32 °C), resulting in controlled drug delivery from the MS@PNIPAm-PAAm system. Furthermore, the 3-(4,5-dimethylthiazol-2-yl)-2,5-diphenyltetrazolium bromide (MTT) assay and cellular internalization results support the prepared MS@PNIPAm-PAAm NPs being biocompatible and readily taken up by MDA-MB-231 cells. The prepared MS@PNIPAm-PAAm NPs, with their pH-responsive drug release behavior and good biocompatibility, could be used as a drug delivery vehicle where sustained drug release at higher temperatures is required.

## 1. Introduction

Cancer has attracted a lot of research focus in the last few decades because of its relationship to the environment and genetics, and it is still one of the major causes of mortality in the twenty-second century [1,2,3]. Chemotherapy has become the mainstay of cancer treatment among traditional cancer treatments (surgical resection, radiation therapy) due to its high efficiency in comparison to other treatments [4]. However, because most chemotherapeutic drugs lack specificity, they kill tumor cells while causing toxic side effects in normal tissues, severely limiting their clinical application. The stimulus-triggered drug carrier system is combined with the stimulus-responsive components and drug, allowing for timed and targeted drug delivery in response to internal/external stimuli [5,6,7,8]. It has the potential to improve drug bioavailability while decreasing the undesirable side effects of chemotherapy drugs [9,10,11].

Stimulus-responsive drug delivery systems are classified as either internal stimuli-responsive (such as pH, temperature, enzyme) or external stimuli-responsive (such as light and magnetic field) [12,13,14,15,16]. However, because the microenvironment of disease sites differs greatly from that of normal tissues, a single stimulus-responsive drug delivery system is unable to meet the demands of modern drug therapy. Multiple stimulus-response materials can help the lesion site adapt to its complex environment and allow for controlled drug release. It has emerged as a critical area of growth for the drug delivery system.

To enrich solid tumors, mesoporous silica nanoparticles could take advantage of the enhanced permeability and retention (EPR) effect. Mesoporous silica materials (MS) are regarded as an ideal candidate for drug delivery applications due to a variety of distinct characteristics when compared to other nanoparticles, including easy synthesis steps, controlled pores, excellent mechanical stability, and good cytocompatibility [17]. MSs’ high surface areas and pore sizes make it easier to load and distribute a significant amount of drug to tumors in a controlled manner. By altering the pore blocking agent, the drug can be placed into the mesopore channel and shielded from premature release. An efficient drug carrier must prevent premature drug leakage before reaching the target sites and release the drugs rapidly at the specific sites [18].

Poly(N-isopropyl acrylamide) (PNIPAM) is a typical example of a temperature-sensitive polymer, with dramatic volume changes at temperatures below the lower critical solution temperature (LCST) of 32 °C [19]. The polymer is in a highly swollen state when the temperature is below LCST. When the temperature rises above LCST, the polymer transforms into a dehydrated, collapsed state, which promotes drug release [20]. Although the LCST of PNIPAM is lower than the physiological temperature of humans, the LCST can be increased by adding hydrophilic copolymer monomers [21]. PAAm, a linear-chain structure polymer with amide substituents on alternating carbon atoms, has long been utilized in contact lenses, and its hydrophilic and bioinert properties make it appealing for biomedical applications [22,23]. As a result, the PNIPAm-PAAm copolymer could be used as a coating in biological applications. To control drug release in response to environmental stimuli, a variety of polymers have been used to coat silica materials [24,25,26,27]. The application of a PNIPAm-PAAm copolymer to the outer surface of MS is anticipated to act as a shell that is pH-responsive to regulate the drug release under the desired pH, which could be useful in controlling the release of chemotherapeutic drugs.

The pH- and temperature-responsive drug delivery system (MS@PNIPAm-PAAm NPs) was created in this study. The temperature-responsive copolymer PNIPAm-PAAm was modified on the MS surface by an epoxy-ring opening reaction approach. To examine the drug release properties, doxorubicin (Dox) was utilized as a model drug. The loaded Dox was successfully protected inside the mesopore channels, allowing it to be released under temperature-stimulated intracellular pH conditions. The cytocompatibility and cell uptake of the prepared MS@PNIPAm-PAAm NPs were assessed using a human breast cancer (MDA-MB-231) cell line.

## 2. Materials and Methods

### 2.1. Chemicals and Reagents

Tetraethyl orthosilicate (TEOS, 99%), cetyltrimethylammonium bromide (CTAB, 99%), N-isopropyl acrylamide (NIPAm, 97%), acrylamide (AAm, 98%), 3-glycidoxypropyl trimethoxysilane (GPTMS, 98%) azobisisobutyronitrile (AIBN, 98%), and doxorubicin hydrochloride (Dox), tetrahydrofuron (THF, 99%), toluene (98%), hydrochloric acid (HCl, 36%), sodium hydroxide (NaOH, 98%), ammonium nitrate (NH_4_NO_3_, 98%) were purchased from Aldrich Chemical Co., Saint Louis, MO, USA. All the chemicals were used as received. Human breast cancer cells, MDA-MB-231 (Korea cell line bank, Seoul, Republic of Korea), were purchased and used for this study.

### 2.2. Synthesis of PNIPAm-Acrylamide (PNIPAm-PAAm) Copolymers

In a typical synthesis of the PNIPAm-PAAm copolymer, NIPAm (1 g, 0.0088 mol) was dissolved in 10 mL THF in a three-neck flask and degassed with nitrogen. To this, acrylamide (0.62 g, 0.0088 mol) was added, and the obtained monomer solution was continuously purged with nitrogen for 45 min before AIBN was introduced to initiate the reaction. Next, the reaction flask was heated to 70 °C for 24 h. After the specified time, the mixture was concentrated, and the resulting sticky, viscous mass was then dissolved in chloroform and reprecipitated from cold hexane [28,29]. The reprecipitation process was done thrice. The obtained copolymer product was dried in a vacuum. We chose NIPAm monomer to incorporate thermoresponsive property and AAm for chemical conjugation onto mesoporous silica nanoparticles through GPTMS linking agent. The synthesized copolymer obtained was a random copolymer with an average molecular mass of about 16,500 g/mol.

### 2.3. Synthesis of Mesoporous Silica (MS) NPs

To dissolve CTAB completely, 1.0 g in 450 mL deionized water was used and allowed to stir at 35 °C. To this, about 5.0 mL of NaOH solution (2.0 M) was added and stirred at 35 °C. The mixture was then stirred at 90 °C after the gradual addition of 5.0 mL of TEOS solution. Finally, the resultant white precipitate was centrifuged, washed several times with water, and dried at 60 °C under vacuum for 24 h. The as-made sample obtained was named MS@CTAB NPs (Figure 1, Step 1) [30].

### 2.4. GPTMS Modification onto the MS@CTAB NPs

To perform this process, MS@CTAB NPs (0.3 g) were suspended in 60 mL of dry toluene. To this, GPTMS (1 mL) was dropped, and the suspension was refluxed at 80 °C for 8 h under inert conditions. The resultant product was centrifuged and washed with toluene (3 × 10 mL each) and subsequently washed with ethanol (3 × 10 mL each) to remove the toluene, and then dried at 60 °C [31]. The obtained GPTMS-modified sample was named MS@CTAB@GPTMS NPs (Figure 1, Step 2). In the next step, the occluded CTAB was extracted by dispersing MS@CTAB@GPTMS NPs (0.2 g) in a 100 mL ethanol solution containing ammonium nitrate (0.3 g) and refluxing for 6 h at 60 °C [32]. The extraction steps were performed thrice and then separated, rinsed with ethanol, and dried at 60 °C. The obtained material was named MS@GPTMS NPs (Figure 1, Step 2).

### 2.5. Dox Loading into the MS@GPTMS NPs

For Dox loading, MS@GPTMS NPs (0.1 g) were dispersed in toluene (10 mL). Next, 5 mL (1 mg/mL) of Dox was introduced and stirred for 24 h. Then, the resultant suspension was centrifuged and rinsed with 1 mL of toluene and a minimum amount of ethanol (1 mL) to eliminate the physisorbed drugs and toluene solvent, and then vacuum dried at 60 °C. The Dox-encapsulated sample was named MS@Dox/GPTMS NPs. The encapsulated Dox was estimated at 498 nm using a UV-Vis spectrometer as follows:$$\mathrm{Drug}\;\mathrm{loading}\;\left(\mathrm{wt}\%\right)=\frac{\mathrm{Mass}\;\mathrm{of}\;\mathrm{the}\;\mathrm{drug}\;\mathrm{in}\;\mathrm{sample}}{\mathrm{Mass}\;\mathrm{of}\;\mathrm{the}\;\mathrm{drug}\;\mathrm{in}\;\mathrm{feed}}\times 100\%$$

A small amount of drug released during the surface modification process was collected, combined with the final drug solution after drug loading, and used to determine the amount of drug loading into the nanoparticles. From the initial and final drug solutions, the loaded Dox was determined to be ~72% (Figure 1, Step 2).

### 2.6. Synthesis of PNIPAM-PAAm Coated MS@Dox/PNIPAm-PAAm NPs

The PNIPAm-PAAm copolymer coated MS@Dox/PNIPAm-PAAm NPs were prepared as follows. To perform this process, MS@GPTMS NPs (0.1 g) were suspended in 50 mL of dry THF. Next, about 2 mL of PNIPAm-PAAm copolymer solution was added to the suspension in the presence of a trimethylamine catalyst and allowed to react at 40–70 °C for 24 h. During this reaction process, the amine (-NH_2_) groups of PAAm can interact with the hydrogen groups of the opened epoxy ring via hydrogen bonding interactions between the amine part of the PAAm segment and the GPTMS units that were modified onto the outer surface of the mesoporous silica nanoparticles (Figure 1, Step 2). Generally, this ring opening reaction requires a high temperature and catalytic conditions to perform a complete reaction [33,34]. However, in this work, an appropriate temperature condition was used to achieve a partial reaction to avoid the crowding accumulation of PNIPAm-PAAm copolymer onto the silica surfaces. The polymer-modified silica sample was collected by filtration, rinsed with THF, and vacuum dried at RT. The obtained PNIPAm-PAAm copolymer modified material was named as MS@Dox/PNIPAm-PAAm NPs.

### 2.7. Stimuli-Responsive In Vitro Drug Release Study

Phosphate buffer saline (PBS) solution was used for the drug release study after being adjusted to the appropriate pH by using 0.1 M HCl or 0.1 M NaOH solution. In vitro Dox release from the MS@Dox/PNIPAm-PAAm NPs was examined under different conditions: (i) different pH (5.0, 6.5, and 7.4); (ii) different temperature (25 °C and 42 °C); (iii) the combined pH and temperature (7.4/25 °C, 5.0/25 °C, and 7.4/42 °C, 5.0/42 °C), respectively. In a dialysis bag (molecular weight cutoff of 10,000), 10 mg of MS@Dox/PNIPAm-PAAm NPs were suspended in PBS (5 mL) at different pHs (7.4, 6.5, and 5.0, respectively). The dialysis bag was immersed in 20 mL of phosphate buffer saline (PBS) medium maintained at the appropriate temperature using a water bath. The samples were then placed in a beaker equipped with a magnetic stirrer. At the predetermined time, 1 mL of the release medium was extracted and analyzed at 498 nm using a UV-Vis spectrophotometer. The control sample MS@Dox/PNIPAm-PAAm NPs was prepared without Dox loading.

### 2.8. Biocompatibility (MTT Assay) Analysis

The in vitro biocompatibility of the generated MS@PNIPAm-PAAm NPs without Dox loading and Dox-loaded MS@Dox/PNIPAm-PAAm NPs, as well as the same concentration of pure Dox, were evaluated on the MDA-MB-231 cell line. Firstly, the cells were cultured in a 96-well plate for 24 h at 37 °C and 5% CO_2_. Secondly, different concentrations of MS@PNIPAm-PAAm NPs, MS@Dox/PNIPAm-PAAm NPs, and free Dox, respectively, were added (5 replicate wells for each sample) and cultured for 24 h. Finally, 20 µL of MTT solution was added and incubated for another 4 h. Next, 200 µL of DMSO was added to each well and gently shaken to replace the existing medium. For each well, the absorbance at 570 nm was measured. The percentage of cell viability was calculated by measuring the absorbance at 570 nm.
$$\mathrm{Cell}\;\mathrm{viability}\;\left(\%\right)=\frac{A_{\mathrm{sample}}-{\;A}_{\mathrm{blank}}}{A_{\mathrm{control}}-{\;A}_{\mathrm{blank}}}\times 100$$
where OD_treated_: cells treated with MS@PNIPAm-PAAm NPs without drug, and MS@Dox/PNIPAm-PAAm NPs, respectively. OD_control_: control cells.

### 2.9. Fluorescence Microscopy Analysis

Fluorescence microscopic examination was used to investigate the intracellular internalization of MS@Dox/PNIPAm-PAAm NPs. For this, the MDA-MB-231 cells were cultured in a 6-well plate in DMEM medium for 24 h. The cultured plate was then incubated for a further 6 h after being treated with MS@Dox/PNIPAm-PAAm NPs at a concentration of 10 µg/mL. Following incubation, the sample-exposed cells were fixed using 4% paraformaldehyde, the existing medium was withdrawn, and the plate was rinsed thrice using fresh PBS buffer to eliminate any adhered samples. Using a fluorescence microscope, fluorescence pictures of the cells were captured.

### 2.10. Instrumental Characterization

Low-angle X-ray diffraction (XRD) investigation was carried out in the low-angle range (2θ = 0.5° to 6°) utilizing a Bruker AXN analyzer. N_2_ sorption analysis was performed by using the Nova 4000e surface area instrument. Fourier transform infrared (FTIR; JASCO FTIR 4100) spectra with a frequency ranging from 4000 to 400 cm^−1^, were recorded using the KBr pellet technique. The surface morphology was studied by scanning electron microscopy (SEM, S4800). Transmission electron microscopy (TEM) images were collected on a JEOL 2010 operating at an acceleration voltage of 200 kV. Scanning electron microscopy (SEM) images were recorded with a JEOL 6400 microscope operating at 20 kV. Thermogravimetric analysis (TGA) was performed on a Perkin-Elmer Pyris Diamond thermogravimetric analyzer with a heating rate of 10 °C/min under N_2_ atm. Particle size and zeta potential measurements were performed on the Malvern Zetasizer Nano-ZS. UV-vis spectral analysis was performed using an Agilent Inc. UV-Vis spectrophotometer.

## 3. Results and Discussion

### 3.1. Synthesis of PNIPAm-PAAm Copolymers, Surface Modification, and Characterization of PNIPAm-PAAm Modified MS@PNIPAm-PAAm NPs

The PNIPAm-PAAm copolymer was synthesized via free radical copolymerization of NIPAm and AAm as shown in Figure 1A, and the obtained copolymer product was characterized by ^1^H NMR and FTIR analysis (Figure 1). Figure 1A displays the ^1^H NMR spectrum of the PNIPAm-PAAm copolymer. As shown in Figure 1A, the proton signals around 1.1 ppm attributed to the methyl (–CH_3_) groups of NIPAm units, and the peaks at 3.7–4.1 ppm are allocated to the –CH_2_-CH_2_O- protons of NIPAm amide groups. The signals in the range of 1.9 to 2.2 ppm and 1.2 to 1.8 ppm are assigned to secondary and tertiary carbon and amine protons in the copolymer backbone, respectively. Figure 1B shows the FTIR spectrum of the synthesized PNIPAm-PAAm copolymer. The strong band appeared at 3425 cm^−1^ which is ascribed to the bending of amine (-NH_2_) groups. The vibration bands at 1637 cm^−1^ and 1715 cm^−1^ are indicating the presence of carbonyl (–C=O) units of acrylamide and NIPAm, respectively. Moreover, the bending vibration band at 2781 cm^−1^, 2922 cm^−1^ indicates the presence of alkyl C-H groups in the PNIPAm in the copolymer backbone. These FTIR absorption peaks demonstrated the successful synthesis of the PNIPAm-PAAm copolymer.

Figure 1C,D depicts the FTIR spectra of MS@GPTMS NPs and MS@PNIPAm-PAAm NPs, respectively. Figure 1C(a) shows that the existence of a propyl carbon chain was indicated by a -C–H band at 2863 cm^−1^, 2936 cm^−1^, and a stretching band at 1389 cm^−1^ was attributed to the C–O vibrations of the epoxy moiety of the GPTMS groups. In addition, the characteristic bands at 963 cm^−1^, 1092 cm^−1^ were identified as vibration peaks of Si-OH and Si-O-Si stretching of MS@GPTMS NPs, respectively. As shown in Figure 1D, after PNIPAm-PAAm copolymer modification onto the MS@GPTMS NPs, the new intense absorption peak of MS@PNIPAm-PAAm NPs appeared at 1639 cm^−1^, 1449 cm^−1^ could be assigned to the amide (-C=O) and (-C-N, -N-H bending) in NIPAm and acrylamide, respectively. As compared to the sample MS@GPTMS NPs, the copolymer-modified MS@PNIPAm-PAAm NPs showed a noticeable peak shift of ~45 cm^−1^ for the C-O peak, indicating the epoxy ring opening by the reaction of epoxy groups with the amine part of the PAAm segments. Moreover, the peak intensity of the alkyl C-H groups was considerably increased, and new peaks appeared for O-H at 1364 cm^−1^, C-N for 1542 cm^−1^, and 1637 cm^−1^ for C=O groups, respectively, which implies that the copolymer has been modified onto the MS@PNIPAm-PAAm NPs. From the FTIR analysis data, the surface modification of the PNIPAm-PAAm copolymer onto the MS@GPTMS NPs was confirmed [35].

The size and morphology of MS@GPTMS NPs and MS@PNIPAm-PAAm NPs were observed by SEM and TEM (Figure 2). As illustrated in Figure 2A, the SEM images of MS@GPTMS NPs have a particle size of about ~175 nm and a uniform pore structure. As displayed in Figure 2b, the size of the particle increased slightly after surface modification, and the particles were slightly aggregated due to the surface modification of the PNIPAm-PAAm polymer onto the MS@GPTMS NPs surface. Meanwhile, the mesopore channels were completely covered by the surface-coated PNIPAm-PAAm copolymer on the MS@PNIPAM-PAAm NPs and are visible and distinguishable when compared to MS@GPTMS NPs, indicating that they were successfully coated with copolymer (Figure 2c,d).

Furthermore, the particle size distribution of MS@GPTMS NPs and MS@PNIPAm-PAAm NPs was determined using a dynamic light scattering (DLS) analysis (Figure 3A). The average particle size of the MS@GPTMS NPs was determined to be between ~75–300 nm, with a polydispersity index (PDI) value of about 0.18, as illustrated in Figure 3A(a). The particle size distribution of the PNIPAm-PAAm copolymer-modified MS@PNIPAm-PAAm NPs, on the other hand, is about ~100 to ~350 nm with a determined polydispersity index (PDI) value of about 0.25 (Figure 3A(b)). The polydispersity value was slightly increased for the MS@PNIPAm-PAAm NPs as compared to the MS@GPTMS NPs, indicating that the particles were monodispersed in an aqueous medium. The presence of a surface-modified PNIPAm-PAAm copolymer on the MS@PNIPAm-PAAm NPs could account for the increased particle size of around ~25–40 nm. Moreover, no considerable small or large particles were observed below ~75 nm and above ~350 nm. However, when increasing the medium temperature to 42 °C, the average particle size was slightly decreased to about ~150 nm due to the shrinkage of surface-modified PNIPAM-PAAm copolymers at higher temperatures. Figure 3B(a,b)) shows the surface charge of MS@GPTMS and MS@PNIPAm-PAAm NPs, respectively. As observed in Figure 3B(a), the MS@GPTMS NPs had a negative zeta potential value of approximately −2.6 mV at pH 5 and ~−32 mV at pH 10. A slight negative zeta potential value was observed for the MS@GPTMS NPs at an acidic pH (pH 5) due to the presence of surface silanol (Si-OH) groups of silica nanoparticles. In contrast, the PNIPAm-PAAm modified MS@PNIPAm-PAAm NPs had more positive values of approximately +24 mV at pH 5 and a negative value of about ~ −30 mV at pH 10 (Figure 3B(b)). As compared to the zeta potential value of MS@GPTMS NPs, the PNIPAm-PAAm copolymer modified MS@PNIPAm-PAAm NPs showed an increased positive zeta potential value due to the presence of -CO-NH- groups in the PNIPAm and PAAm segments which undergo protonation (-CO-NH^+^) under an acidic pH condition. Therefore, the overall zeta potential value for MS@PNIPAm-PAAm NPs showed a positive charge under the low pH condition. These results indicate that the PNIPAm-PAAm copolymer has been modified onto the mesoporous silica MS@GPTMS NPs.

The XRD patterns of the MS@GPTMS NPs showed three crystal diffraction peaks at 2θ = 2.13°, 3.62°, and 4.19°, which were assigned to (100), (110), and (200), respectively, indicating the formation of ordered mesoporous materials (Figure 4A(a)). However, after PNIPAm-PAAm copolymer modification, the diffraction crystal plane at 2θ = 1.75° (100) shifted toward a lower angle and the peak intensity decreased, and the peaks at 2θ = 3.62° and 4.13° (110) and (200) are not visible. This might be the ratio of ordered material: the total sample decreased, and therefore, the diffraction planes became less visible due to the presence of surface-modified PNIPAm-PAAm copolymers on the MS@PNIPAm-PAAm NPs. This result indicates that the PNIPAm-PAAm copolymer was covered onto the mesopore channels of the MS@PNIPAm-PAAm NPs [36]. 

N_2_ adsorption-desorption analysis was used to examine the surface and mesopore properties of the MS@GPTMS and MS@PNIPAm-PAAm NPs. The specific surface area was estimated by the Brunauer–Emmet–Teller (BET) method. The pore size distribution curve was obtained from an analysis of the adsorption branch using the Barret–Joyner–Halenda (BJH) method. Both samples had type IV isotherms, as illustrated in Figure 4B(a,b), indicating that the presence of mesoporous structures in the prepared mesoporous silica nanoparicles. The MS@GPTMS NPs’ surface area, mesopore size, and pore volume were calculated to be about ~658 m^2^/g, 2.3 nm, and 0.98 cm^3^/g, respectively. The MS@PNIPAm-PAAm NPs, on the other hand, had a significantly reduced surface area of ~152 m^2^/g after PNIPAm-PAAm copolymer modification (Figure 4B). The presence of surface coated PNIPAm-PAAm copolymers on the MS@PNIPAm-PAAm NPs resulted in a significant reduction in surface area and pore size (Figure 4C). These results also supported the finding that the PNIPAm-PAAm copolymer was successfully modified on the surface of MS@GPTMS NPs.

Figure 5 depicts the TGA curves for the GPTMS modified sample, MS@GPTMS NPs, MS@GPTMS NPs without the CTAB extraction process, and PNIPAM-PAAm copolymer coated MS@PNIPAm-PAAm NPs, respectively. The MS@GPTMS NPs showed about 11.3% indicating the degradation of surface-modified GPTMS functional units. Further, the MS@GPTMS-CTAB sample showed ~18.5% weight loss due to the degradation of occluded CTAB molecules. The MS@PNIPAm-PAAm NPs had an initial weight loss of about ~2.5 wt.% between 30 °C and 100 °C due to the evaporation of physisorbed water or moisture. Moreover, the PNIPAM-PAAm copolymer-coated MS@PNIPAm-PAAm NPs showed a gradual weight loss of about ~24.5 wt.% in mass, which was implied to be due to the thermal degradation of surface-modified PNIPAm-PAAm copolymer from the surface of the mesoporous silica nanoparticles.

### 3.2. Temperature-Responsive Dox Release from MS@Dox/PNIPAM-PAAm NPs

PNIPAm is a prominent thermoresponsive polymer composed of hydrophilic amide and hydrophobic alkyl groups, with an LCST of around 32 °C, which is somewhat lower than the body temperature (37 °C). As a result, PNIPAm solution has a sol state at ambient temperature and a gel state when heated to body temperature. This distinct characteristic enables PNIPAm to be used in biological applications. Similarly, the PAAm polymer features amine (-NH) groups in the side chain that can serve as drug binding sites to hold the drugs under physiological pH circumstances and release the loaded drugs under acidic pH conditions owing to protonation of -NH- sites. The combination of thermoresponsive and pH-responsive characteristics of the synthesized PNIPAm-PAAm copolymer make it an excellent candidate for the modification of silica nanoparticles for drug delivery applications. Temperature-responsive drug release behavior of the Dox-loaded MS@Dox/PNIPAm-PAAm NPs system was evaluated at different temperatures of 25 °C and 42 °C, respectively (Figure 6a). As shown in Figure 6a, only a small amount of Dox release, ~17.6%, was seen in the 48-h release period. These results revealed that the H-bonding interactions between the -C=O and -N-H groups present in the PNIPAm-PAAm copolymer are more stable at pH 7.4 and 25 °C, implying that the PNIPAm-PAAm copolymer on the MS@Dox/PNIPAm-PAAm NPs effectively retains the drugs from premature release at pH 7.4 and at 25 °C conditions. 

In contrast, due to the temperature-responsive shrinkage of PNIPAm segments at 42 °C, the Dox release was increased to ~69.8% even at pH 7.4 (Table 1). As a result, the pore channels were opened, and the loaded drugs began to release from the mesopore channels. As shown in Figure 6a, as the temperature increased, more drugs were released from MS@Dox/PNIPAm-PAAm NPs, indicating that the prepared MS@Dox/PNIPAm-PAAm NPs is temperature responsive. The main reason was that when the temperature fell below LCST (<32 °C), the copolymer layer on the drug carrier’s surface swelled, and therefore, the Dox was blocked in the pore channel, reducing the release. When the temperature exceeded LCST (>32 °C), the surface-modified PNIPAm-PAAm polymer layer on the carrier surface collapsed, exposing the pore channel and promoting the release of large amounts of drug from MS@Dox/PNIPAM-PAAm NPs [25,37].

### 3.3. Stimuli-Responsive Dox Release from MS@Dox/PNIPAm-PAAm NPs at Different pH Conditions

Drug release behavior of the MS@Dox/PNIPAm-PAAm NPs was accessed under different pH (pH 7.4 and 5.0) stimuli conditions, respectively. Figure 6b shows the cumulative Dox release at pH 7.4 was 18.5% after 48 h, while at pH 5.0, the cumulative release was increased to ~63.5% after 48 h. At pH 7.4, only a small amount of Dox (about ~18.5%) was released during a 48-h period, indicating that surface-coated PNIPAm-PAAm copolymers retain the drug molecules inside mesoporous carriers due to the existence of H-bonding interactions between the –C=O and -NH groups of PNIPAm and PAAm, resulting in less drug release. These phenomena could be explained by using zeta potential measurements. As shown in Figure 3B, at pH 7.4, the MS@PNIPAm-PAAm NPs had a higher negative charge, indicating that the loaded Dox molecules interacted significantly with drug binding sites such as hydroxyl and amine sites. As a result, it demonstrated a significantly lower level of Dox release induced by the release of physisorbed drug molecules. However, when the pH was decreased to pH 5.0, the -NH and -OH groups, as well as the surface hydroxyl groups, became protonated under the reduced pH (pH 5.0) conditions. According to the zeta potential measurements, the MS@PNIPAM-PAAm NPs became more positively charged under the acidic pH conditions (pH 5.0), and thus the amine groups in the PNIPAm and PAAm copolymer and hydroxyl groups in silica surfaces were protonated to -NH^+^ and -OH^+^, resulting in a higher cumulative release of Dox at pH 5.0 than the release of Dox at pH 6.5 due to an increased electrostatic repulsive force between protonated Dox molecules and the protonated drug binding sites. With decreasing pH, the protonation degree of -NH^+^ increased, resulting in greater electrostatic repulsion between the copolymer chains. As a result, after 48 h, the cumulative Dox drug release at pH 5.0 was ~63.5% and about 3.5 times higher than at physiological pH (7.4), confirming the MS@Dox/PNIPAm-PAAm NP’s excellent pH responsiveness (Figure 6b and Table 2) [38].

### 3.4. Combined pH and Temperature-Responsive Dox Release from MS@Dox/PNIPAm-PAAm NPs

The Dox release from the MS@Dox/PNIPAm-PAAm NPs at pH 7.4/25 °C, 7.4/42 °C, and 5.0/25 °C and 5.0/42 °C, respectively, are shown in Figure 6c. At pH 7.4/25 °C, approximately ~35% of the Dox was released, and at pH 7.4/42 °C, Dox release increased to about ~81.5% as shown in Figure 6c. In contrast, at lower pH such as pH 5.0/25 °C and pH 5.0/42 °C, the MS@Dox/PNIPAm-PAAm system released ~67% and complete release of Dox, respectively (Figure 6c and Table 2). In comparison to Dox release only under acidic pH or temperature circumstances, the proportion of Dox released under the combined acidic pH and temperature (42 °C) stimulus conditions was significantly higher. This suggests that the combination of pH and temperature is the key driving factor in the release of drugs from the mesopore channels. Protonation of -NH groups and dissociation of H-bonding interactions between PNIPAm-PAAm copolymer chains can be caused by acidic pH stimuli. Furthermore, at temperatures above LCST, the PNIPAm segments shorten and open pores, and the loaded drugs are released quickly from the mesopore channels. As a result, Dox release was significantly higher at pH 5.0/42 °C than at either pH or temperature alone [39]. In this study, it was found that at pH 7.4, only ~18.5% of the drug was released in 24 h, implying that the loaded drug was firmly confined inside the mesopores under these pH 7.4 conditions. However, at higher temperatures (25 °C and 42 °C), around ~17.6 and ~64.8% were released in 24 h due to slow dissolution and diffusion out of the mesopore channels, respectively. The drug carrier system requires additional time to achieve complete release (over 24 h). In contrast, when the acidic pH (pH 5.0) and temperature (42 °C) stimuli were combined, an enhanced drug release and approximately complete drug release were achieved in 24 h. This might be attributed to the acidic pH-induced electrostatic repulsion as well as the temperature-induced drug solubility and fast kinetics, which all play important roles in boosting drug release from the drug carrier system, and therefore nearly complete release was achieved in 24 h.

The release of Dox can be studied by altering the condition from pH 7.4/25 °C to pH 5.0/42 °C to access the MS@Dox/PNIPAm-PAAm NPs. For the first 12 h, the pH was kept at 7.4 at 25 °C, and the Dox release was studied. Then, the pH and temperature were changed to pH 5.0 after 12 h, and the released drug was measured for 48 h. Only a small percentage (~19.5%) of Dox was released over the first 12 h, as shown in Figure 6d. When the pH was raised to 5.0/42 °C, however, it resulted in a ~95% increase in Dox release (Figure 6d). In this study, a slightly increased ~32% of Dox release was observed at pH 5.0 at 42 °C compared to only at pH 5.0 (Table 2). The combined pH and temperature stimulation has been shown to significantly increase drug release behavior, and therefore the obtained results evidence the dual pH and temperature responsiveness of the prepared MS@PNIPAm-PAAm NPs system [40].

### 3.5. Cytotoxicity of MS@PNIPAm-PAAm NPs

Biocompatible polymers or functional groups of modified silica materials showed considerable cytocompatibility [41,42]. To determine the in vitro biocompatibility, the PNIPAm-PAAm copolymer modified MS@PNIPAm-PAAm NPs were tested at the sample doses of 0, 25, 50, 100, 150, and 200 µg/mL, respectively, without or with Dox loading using MDA-MB-231 cells. The cell viability of MS@PNIPAm-PAAm NPs without Dox loading had ~90% cell viability in all the tested concentrations, as shown in Figure 7A, indicating that MS@PNIPAm-PAAm NPs have favorable biocompatibility. Further, the cytotoxicity of free Dox and Dox-loaded MS@Dox/PNIPAm-PAAm NPs, respectively, was investigated. All of the samples displayed concentration-dependent cell toxicity to MDA-MB-231 cells, as shown in Figure 7B. Specifically, the Dox-loaded MS@Dox/PNIPAm-PAAm NPs-treated cells showed considerably enhanced cell toxicity of ~>95% to the MDA-MB-231 cells treated with the MS@Dox/PNIPAm-PAAm NPs concentration of about ~200 μg/mL after being cultured for 24 h at 37 °C (Figure 7B(b)).

As shown in Figure 7B(b), there was a significant increase in toxicity at the sample concentration of 200 μg/mL as compared to the sample concentration of 25 μg/mL or 50 μg/mL, indicating that more drug molecules were released from the MS@Dox/PNIPAm-PAAm NPs system due to the increased cellular uptake of the nanoparticles and the intracellular pH conditions allowing the mesopores to open and allow the drugs to be released from the mesoporous silica carriers. MS@Dox/PNIPAm-PAAm NPs showed considerably more toxicity to the MDA-MB-231 cells than the cells treated with the same concentration of free DOX (Figure 7B(c)). This study’s results indicate that the prepared MS@PNIPAm-PAAm NPs are biocompatible, and Dox-loaded MS@Dox/PNIPAm-PAAm NPs are more easily endocytosed by cells, resulting in increased intracellular Dox concentration and increased tumor cell killing efficiency.

### 3.6. Cell Uptake Study

To assess intracellular uptake, MS@Dox/PNIPAm-PAAm NPs were treated for 5 h with MDA-MB-231 cells at a sample dose of 10 μg/mL at 37 °C. Figure 8 shows the cellular uptake and consequent internalization of the MS@Dox/PNIPAm-PAAm NPs into the MDA-MB-231 cells are observed by the Dox fluorescence, demonstrating that the MS@Dox/PNIPAm-PAAm NPs can be engulfed by MDA-MB-231 cells in a short incubation time (Figure 8b,c). This could be due to the surface-coated PNIPAm-PAAm copolymer, which promotes cellular internalization of MS@Dox/PNIPAm-PAAm NPs by MDA-MB-231 cells and thus facilitates drug release in a controlled manner (Figure 8b,c). The intracellular uptake behavior of the MDA-MB-231 cells was also examined at different time (1, 2, 4, and 8 h) intervals. According to Figure 8d, increased fluorescence intensity of Dox in the cells with respect to time indicates effective cellular internalization of MS@Dox/PNIPAm-PAAm NPs inside the cells via the intracellular internalization process [43,44,45]. The fluorescence image results provide direct evidence for the pH-responsive delivery performance of Dox from the MS@Dox/PNIPAm-PAAm NPs.

## 4. Conclusions

In this study, PNIPAm-PAAm copolymer conjugated mesoporous silica MS@PNIPAm-PAAm NPs were synthesized for pH- and temperature-responsive in vitro drug delivery applications. The surface-conjugated PNIPAm-PAAm copolymer acts as a gatekeeper to prevent the drugs from impulsive release under physiological conditions, while the mesoporous silica serves as a drug reservoir. The MS@Dox/PNIPAm-PAAm NPs demonstrated improved drug release performance under pH and temperature stimuli due to the pH-responsive collapse and temperature-responsive shrinkage of the surface-conjugated PNIPAm-PAAm copolymer above the LCST. The MS@PNIPAm-PAAm system is biocompatible and readily engulfed by MDA-MB-231 cells, according to the in vitro biocompatibility (MTT assay) results. All the experimental results support the idea that the MS@PNIPAm-PAAm NPs could be used for pH- and temperature-responsive drug delivery that requires the target sites to be exposed to sustained drug release at higher temperatures.

## Data Availability

Not applicable.

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
