# Peer review of "Thermosensitive Polymer-Modified Mesoporous Silica for pH and Temperature-Responsive Drug Delivery"

_pharmaceutics, 2023, doi:10.3390/pharmaceutics15030795_

Round 1
Reviewer 1 Report
The manuscript submitted by Thirupathi et al. describes the synthesis of MCM-41 nanoparticles, its functionalization with epoxy groups by grafting, and covering with a thermoresponsive polymer PNIPAm-PAAm, aiming the development of a stimuli-responsive drug delivery system. Doxorubicin was used as a model drug. The work is quite interesting, as doxorubicin release can be triggered by pH and temperature, and the maximum release rate was achieved at pH 5.0, matching the pH of cancer cells. In order to be accepted in Pharmaceutics, however, several queries must be resolved. Therefore, my recommendation is for major revision.
-
FTIR data: FTIR spectrum is not clear evidence of PNIPAm modification onto MS@GPTMS surface, as both spectra are quite similar. In the spectrum, where is the band of C-O stretching discussed in the text? It seems that a small shift in bands between 1400 and 1800 are observed. The data must be better discussed. Moreover, Fig 1c lacks a -1 in the x-axis.
-
DLS data: what is the polydispersity index of the measurement? This data must be included in order to ensure the quality of the measurement. Additionally, authors should comment if no bigger or smaller particles were observed, as the scale is cutted in the x-axis.
-
Zeta potential data: authors must discuss the charge of the nanoparticles accordingly with the chemical structure of the polymers. Moreover, why does MS@GPTMS have positive surface charge?
- DRX data: the lack of peaks in the diffractogram does not imply that silica lost its pore orderliness, as this would go on contrary to the mechanical stability of silica. Most likely the ratio ordered material:total sample decreased and the diffraction planes became less visible/accessible. Please, discuss.
- Sorption analysis data: In pore size distribution, how does the authors estimate the size of 1.1 nm for polymer-covered MS? From the graph, it is not possible to estimate the pore size because there is no pore size distribution curve, as pores were blocked. Moreover, what model was used to fit the pore size distribution: BJH or DFT? Please, clarify.
- TGA data: authors must provide TGA measurements of MS@GPTMS to correctly associate the organic moieties with the weight loss regions. Ideally, authors should also provide TGA measurement of MS@GPTMS loaded with CTAB to check if there is no CTAB residual contribution.
- In vitro drug release as function of pH: discussion should be integrated with zeta potential data.
Minor issues:
-
line 48-49: all stimuli are described as internal.
-
section 2.4: was silica dried previously to the grafting reaction? how did the authors assess the complete removal of CTAB? line 124: atmosphere is abbreviated as atm, but atm is a unity of pressure. Do not abbreviate.
-
section 2.5: toluene has a high boiling point and is not easily removed by vacuum drying at room temperature. How do the authors ensure that toluene was completely removed? If residual toluene is present, this could interfere in the cytotoxicity assay. The equation for drug loading efficiency estimation is not clear. What is the weight of the sample? This process must be clarified.
-
section 2.7: in vitro drug release: sink conditions were considered? Please include information.
-
section 2.10: SEM and TEM analysis are not described. How samples were prepared for FTIR analysis: KBr pellets? ATR crystal apparatus? DLS and zeta potential are described twice with different instruments. Please clarify.
Reviewer 2 Report
This manuscript from Thirupathi et al. is reporting a good work about mesoporous silica-based drug delivery system (MS@PNIPAm-PAAm NPs) was synthesized by conjugating PNIPAm-PAAm copolymer onto the mesoporous silica (MS) surface as a gate-keeper that responds to temperature and pH changes. The prepared MS@PNIPAm-PAAm NPs with its pH-responsive drug release behavior and good biocompatibility. All the results suggested that NPs could be used as a drug delivery vehicle where sustained drug release at higher temperatures is required. The research is interesting, the experimental content is reasonable, the workload is very substantial, and the logic is reliable. I would like to recommend the publication of this work after revision. However, the authors should solve the following issues.
The issues in the manuscript are listed as below:
1. Introduction, the design of pH responsive drug delivery system based on mesoporous silica nanoparticles had been investigated wildly in the field of biomedical, and some relative works must be cited to enrich the background of this paper. Please see references (Microporous and Mesoporous Materials 329 (2022)111512; Materials 15(2022)5926; Journal of Controlled Release 338 (2021) 719-730; Chinese Chemical Letters 32 (2021) 3696–3704).
2. For Figure 6, The accumulative release amount of DOX seems to have already reached the plateau within 24 hours. Why cannot the drug be completely released? Please explain.
3. The in vivo antitumor efficacy should be evaluated, if possible?
4. Overall the paper is easily read, but there are still some minor problems in language and spelling. A throughout checking is recommended.
Reviewer 3 Report
The paper presents interesting results but can be accepted for publication only after major revision. Important issues must be previously clarified.
The reaction between epoxide ring and amide groups can be conducted only in harsh conditions (high temperatures and catalysts). Please note that cited here papers (Chang, K.-C.; Lin, C.-Y.; Lin, H.-F.; Chiou, S.-C.; Huang, W.-C.; Yeh, J.-M.; Yang, J.-C. Thermally and mechanically enhanced epoxy resin-silica hybrid materials containing primary amine-modified silica nanoparticles. J. Appl. Polym. Sci., 2008, 108, 1629- 505 1635. 28. Wach, A.; Drozdek, M.; Dudek, B.; Biazik, M.; Łątka, P.; Michalik, M.; Kustrowski, P. Differences in Catalytic Activity of Poly(vinylamine) Introduced on Surface of Mesoporous SBA-15 by Grafting from and Grafting onto Methods in Knoevenagel Condensation, J. Phys. Chem. C, 2015, 119, 19954−19966.) described interactions with primary amine groups but not amide groups with other chemical properties. At least an appropriate discussion should be provided.
Scheme 1 should be revised. Why are two NIPAM units depicted? It is completely unclear from the scheme whether it is a block copolymer or random. It seems that red and green lines are attached to the surface in parallel. Why did the authors select these monomers for synthesis?
The synthesized polymer is poorly characterized. Is it a block or a random structure? What is molecular mass?
The authors uploaded the Dox before the synthesis of the grafted coatings. What about the release of Dox at the time of the synthesis? Appropriate discussion should be provided.
Subsection 3.1. "Synthesis of PNIPAm-PAAm copolymer" includes information not only about the synthesis of the polymers but also the modification and characterization of the nanoparticles. The title of the subsection should be corrected.
There is a lack of information about molecular mechanisms, thermo- and pH-responsivity, and this should be addressed.
The authors provided a DSC analysis for one pH, but temperature conditions were not suggested. It will be valuable to add information about the behavior of functionalized particles at various pH levels and temperatures. Information about buffer solutions for pH studies is absent from the paper.
Finally, I recommend citing papers where similar systems were intensively studied:
https://doi.org/10.3390/polym14194245
https://doi.org/10.1007/s00396-020-04750-0
Round 2
Reviewer 1 Report
The authors have addressed almost every question. However, there are still some issues to be corrected.
Experimental section:
1. drug loading efficiency's equation still does not make mathematical sense.
2. SEM and TEM equipments and analysis conditions are still not mentioned.
3. The use of KBr pellets in FTIR analysis are still not mentioned.
Results and discussion section:
1. At pH 5.0, silica could not present positive charge, as the Point of Zero Charge (PZC) of silica is well-known as 2-3. At pH 5.0 silica presents NEGATIVE CHARGE, which is the opposite of what the authors discuss. Additionally, it is expected that most of the silanol groups of silica already reacted with GPTMS during grafting reaction. Zeta potential discussion must be revised with proper literature and scientific soundness and accuracy.
Author Response
Reviewer-1:
The authors have addressed almost every question. However, there are still some issues to be corrected.
Response: The authors are grateful to all the reviewers for their valuable time to review our manuscript and their positive recommendation for acceptance.
Specific comments:
Experimental section:
Comment-1: drug loading efficiency's equation still does not make mathematical sense.
Response: We are sorry for the mistake. The specified mistake was corrected in Page 4, Line 154-156 in the revised manuscript.
Comment-2: SEM and TEM equipment’s and analysis conditions are still not mentioned.
Response: Thank you for the kind suggestion. The specified details were included in Page 5, Line 231-233 in the revised manuscript.
Comment-3: The use of KBr pellets in FTIR analysis are still not mentioned.
Response: Thank you for the kind suggestion. The specified details were included in Page 5, Line 228-230 in the revised manuscript.
Results and discussion section:
Comment-4: At pH 5.0, silica could not present positive charge, as the Point of Zero Charge (PZC) of silica is well-known as 2-3. At pH 5.0 silica presents NEGATIVE CHARGE, which is the opposite of what the authors discuss. Additionally, it is expected that most of the silanol groups of silica already reacted with GPTMS during grafting reaction. Zeta potential discussion must be revised with proper literature and scientific soundness and accuracy.
Response: Thank you for the reviewers’ kind suggestion. To address this comment, we have remeasured the sample at acidic pH (pH 5.0) condition and the obtained data has been presented in Page 8, Figure 3(B(a)) and the appropriate discussion has been revised in Page 8, Line 317-330 in the revised manuscript.
Reviewer 3 Report
After revision, the quality of the paper was strongly improved, but at least two important issues should be clarified.
First, it is unlikely that the amide groups of the AAm are able to interact with the epoxide ring with enough efficiency. The authors continue to assert that the chemical properties of the amide and amine groups are the same. An alternative version should be provided in addition. For example, hydrogen bonds between the AAM's amide groups and the opened epoxide rings.
Second, the authors noted that the synthesized copolymer was a block copolymer, which is unlikely. I don't see any reason for it. In some cases, the block copolymers produced by conventional radical polymerization can be synthesized for specific reactivity ratios of the monomers. Please add appropriate discussion.
Author Response
Reviewer-3:
After revision, the quality of the paper was strongly improved, but at least two important issues should be clarified.
Response: The authors are thankful for the reviewers’ positive comment. The specified revisions have been carried out as follows.
Comment-1: First, it is unlikely that the amide groups of the AAm are able to interact with the epoxide ring with enough efficiency. The authors continue to assert that the chemical properties of the amide and amine groups are the same. An alternative version should be provided in addition. For example, hydrogen bonds between the AAM's amide groups and the opened epoxide rings.
Response: Thank you for the reviewers’ kind comment. As suggested, the specified discussion was appropriately revised in Page 4, Line 169-172 in the revised manuscript.
Comment-2: Second, the authors noted that the synthesized copolymer was a block copolymer, which is unlikely. I don't see any reason for it. In some cases, the block copolymers produced by conventional radical polymerization can be synthesized for specific reactivity ratios of the monomers. Please add appropriate discussion.
Response: Thank you for the reviewers’ kind suggestion. The specified discussion was revised appropriately in Page 3, Line 114-115 in the revised manuscript.